# Maternal and child health intervention to promote behaviour change: a population-level cluster-randomised controlled trial in Honduras

William Oles ![ORCID],[1] Marcus Alexander ![ORCID],[1] Rennie Negron,[2] Jennifer Nelson,[3] Emma Iriarte,[3] Edoardo M. Airoldi,[4,5] Nicholas A. Christakis ![ORCID],[1,6] Laura Forastiere ![ORCID] [1,7]

WO and MA contributed equally.

For numbered affiliations see end of article.

**Correspondence to**
Dr Laura Forastiere;
laura.forastiere@yale.edu

## ABSTRACT

**Objectives** To assess the efficacy of a sustained educational intervention to affect diverse outcomes across the pregnancy and infancy timeline.

**Setting** A multi-arm cluster-randomised controlled trial in 99 villages in Honduras' Copán region, involving 16 301 people in 5633 households from October 2015 to December 2019.

**Participants** Residents aged 12 and older were eligible. A photographic census involved 93% of the population, with 13 881 and 10 263 individuals completing baseline and endline surveys, respectively.

**Intervention** 22-month household-based counselling intervention aiming to improve practices, knowledge and attitudes related to maternal, neonatal and child health.

**Primary and secondary outcome measures** Primary outcomes were prenatal/postnatal care behaviours, facility births, exclusive breast feeding, parental involvement, treatment of diarrhoea and respiratory illness, reproductive health, and gender/reproductive norms. Secondary outcomes were knowledge and attitudes related to the primary outcomes.

**Results** Parents targeted for the intervention were 16.4% (95% CI 3.1%–29.8%, p=0.016) more likely to have their newborn's health checked in a health facility within 3 days of birth; 19.6% (95% CI 4.2%–35.1%, p=0.013) more likely to not wrap a fajero around the umbilical cord in the first week after birth; and 8.9% (95% CI 0.3%–17.5%, p=0.043) more likely to report that the mother breast fed immediately after birth. Changes in knowledge and attitudes related to these primary outcomes were also observed. We found no significant effect on various other practices.

**Conclusion** A sustained counselling intervention delivered in the home setting by community health workers can meaningfully change practices, knowledge and attitudes related to proper newborn care following birth, including professional care-seeking, umbilical cord care and breast feeding.

**Trial registration number** NCT02694679.

## STRENGTHS AND LIMITATIONS OF THIS STUDY

⇒ High-quality data acquisition over 4 years using a modified version of Timed and Targeted Counselling methodology included dialogue counselling methods and interactive story-telling to assess current needs and practices of intervention recipients and negotiate on progressive improvement.

⇒ A wider array of outcomes was collected than other trials of its kind, determining the impact of the intervention on various points of the pregnancy and infancy timeline.

⇒ Two-stage randomisation and different treatment dosages allowed estimation of pooled spillover effects on targeted individuals or households, a feature uncommon to other large-scale interventions.

⇒ Limitations include the potential influence of cointervention in the study area; the reliance on self-reported outcomes; and that the intervention modules were delivered at different times and frequencies, depending on household composition.

## INTRODUCTION

Despite significant global progress in many measures of maternal and under-5 child health over the past 30 years, maternal and child morbidity and mortality continue to pose a significant burden, particularly in resource-limited settings across sub-Saharan Africa and Latin America. Globally, neonatal deaths account for nearly 50% of all deaths in children under 5 years old, and preterm birth complications, pneumonia and intrapartum-related complications account for the majority of these deaths.[1 2] These causes suggest that poor maternal outcomes often underlie poor infant and child outcomes; consequently, large-scale interventions to improve resources, knowledge and practices at every point in the pregnancy and infancy timeline have become a central part of the global health agenda. Interventions targeting maternal and child health outcomes are largely founded on evidence-based WHO guidelines, which emphasise birth

preparedness, complication readiness and care-seeking for the mother, as well as immediate stimulation, skin-to-skin contact, immediate and exclusive breast feeding, umbilical cord care, thermal care and medical check-ups for the newborn, among others.[3 4]

Although many neonatal, child and maternal deaths can be prevented through clinical services, emerging evidence has shown that reductions in poor health outcomes in these areas are possible by complementing them with low-cost interventions in the home and community setting.[5 6] This branch of intervention packages focuses on demand-side conditions, building on the idea that improvement in behaviours related to maternal and child health outcomes not only depends on the accessibility or provision of health services, but also on the community-level demand for services and practices. Behaviours which affect maternal and child health outcomes are often socially reinforced or otherwise influenced by community norms and therefore difficult to change, making the on-ground implementation of interventions increasingly important. There is a large body of evidence that points to the efficacy of counselling interventions, which (typically) employ interpersonal interactions with community health workers (CHWs) during home visits and may be implemented on their own or as one of many domains in a multidimensional intervention package.[6–10]

Despite the considerable progress that Honduras has made towards its population health goals, it still lags behind many other low-to-middle-income countries in Latin America. In Honduras in 2016, the under-5 mortality rate was 19.8 per 1000 live births while the neonatal mortality rate was for 11.4 per 1000 live births.[11] Despite important progress, these rates were greater than comparable countries in the same year, including Mexico, Nicaragua and El Salvador, with only Guatemala exceeding that of Honduras.[11] The leading causes of neonatal mortality in Honduras were perinatal disorders, congenital malformations, pneumonia, diarrhoea and malnutrition.[12] With regard to women's and reproductive health, the most recent Honduras Demographic and Health Survey (DHS) reports that the total fertility rate of women aged 15–49 is 2.9, the percent of pregnancy or motherhood among those aged 15–19 is 24%, the median age of first marriage is 19.3 years, and the percent of women who are literate is 92.7.[13] Additionally, in 2020, the WHO estimated that Honduras had a maternal mortality ratio of 72 deaths per 100 000 births, which corresponded to 4.2% of all deaths among women aged 15–49 being attributed to maternal causes.[14] Nationwide, 74% of live births occur in a medical facility, while 57% of live births occur in rural regions which likely face higher barriers to perinatal care resource access, given that Honduras already has the lowest physician density in Latin America.[15–17]

Few studies have combined an at-scale counselling intervention with randomised assessment methods in Latin America. Here, we completed a randomised trial in the western highlands of Honduras in which households received a 2-year intervention package that consisted of household visits by CHWs. CHWs were trained to use validated counselling methods to promote practices, knowledges and attitudes related to desired maternal and child health outcomes. We assessed a wide range of outcomes across the pregnancy and infancy timeline that aligned with the intervention counselling topics, including prenatal care, facility-based births, postnatal care for mother and child, danger signs and care-seeking, breast feeding, paternal involvement, diarrhoea and respiratory illness prevention and management in children 5 and under, and reproductive health. Pertinently, the underlying framework of the intervention here also differed from similar efforts in other settings in that it targeted other members of the community in addition to parents themselves.

The aim of our randomised assessment is thus to evaluate the effects of a large-scale, household-based counselling intervention on outcomes related to maternal, neonatal and child health in rural villages in Honduras.

## METHODS
### Study design and participants
We conducted a multi-arm cluster-randomised controlled trial (RCT) in the Department of Copán in western Honduras comprising the municipalities of Copán Ruinas, Santa Rita, Cabañas and San Jerónimo. Copán is a predominantly rural, mountainous, coffee-growing region and is characterised by high rates of neonatal and maternal morbidity and mortality, with many rural villages facing barriers to healthcare which often leave perinatal care insufficient or unsafe.[12 16] The trial was a partnership between the Yale Human Nature Lab, the Inter-American Development Bank (IDB) and the Ministry of Health of Honduras, with implementing partners World Vision Honduras and Dimagi. Of the 238 villages in the Copán region, 176 were selected for a parent trial evaluating a novel social network targeting technique; of these, 99 were included in the current assessment (villages in which participants were assigned to treatment based on a non-random, network-based algorithm were excluded). The parent trial was formally registered prior to implementation, and the full details of the parent trial design and methods are published in our 2017 study protocol.[17] Factors such as population size, geographic diversity, accessibility and safety were considered when selecting villages, and power testing using conservative treatment-effect simulations following initial enrolment found recruitment to be adequate to detect the desired effects. The village selection process, study area profiling and power testing are described further in our study protocol.[17]

All individuals aged 12 years or older and who lived in a study village were eligible to participate. We conducted a complete photographic census among individuals who agreed to enrol in the study which covered 93% of the eligible population. Any household with at least one eligible individual who agreed to enrol in the study

and who completed the baseline survey was eligible for randomisation.

## Randomisation and masking

The trial used a two-stage factorial design with a first-stage randomisation at the village level and a second-stage randomisation at the household level. In the first stage, we randomly assigned villages to a dosage, or proportion of households targeted for intervention per village (0, 0.05, 0.1, 0.2, 0.3, 0.5, 0.75, 1). In the second stage, households were randomly assigned to the intervention according to the village's dosage. We used a covariate-constrained randomisation procedure that ensured balance at the village and household levels between arms. Our full randomisation procedure is described in our published protocol.[17]

The current randomised assessment focuses on the 99 villages that were assigned to a proportion of targeted households greater than 0 and a random targeting strategy or the control arm with a proportion equal to 0. In other words, we had two kinds of control subjects: (1) untreated households in untreated villages and (2) untreated households in treated villages. This is important because, when assessing whether subjects respond to receiving an educational intervention, one should adjust for whether, by chance, other members of their village also got the same intervention, thereby potentially reinforcing the impact of the educational intervention. We note that this important detail of possible reinforcement of treatment response due to treatment of other community members is typically overlooked in most field trials of such interventions. Households in the 22 villages with a proportion of households targeted equal to 0 did not receive any intervention and were used as our pure control, or comparison, villages.

The intervention assessment reported here establishes if there were changes in practices, knowledge and attitudes related to target outcomes among those actually receiving the intervention. Due to the nature of the intervention, masking of participants and CHWs, termed Community Change Agents (CCAs) in the current study, who delivered the intervention was not possible. However, both CCAs and surveyors were blinded to the methods used to select the intervention households assigned to them.

## Procedures

We and our local partners introduced the project to village leaders, secured local approvals and managed local implementation of the study and intervention. There was extensive local involvement in setting the agenda for this RCT. CCAs delivered up to 22 1–2 hours counselling sessions across 15 modules to targeted households at monthly intervals between November 2016 and August 2018. The intervention was designed by World Vision Honduras (who also hired, paid and trained CCAs) and was named Proyecto Redes: Con Amor y Cuidados Madres y Bebés Sanos (With Love and Care, Healthy Moms and Babies).[18 19] Intervention delivery and outcomes assessment were conducted by two different, independent teams of people.

CCAs made home visits and spoke to families regarding several health topics tailored to the family's current circumstances and based on a modified version of the Timed and Targeted Counselling (ttC) methodology, which has been implemented in 20 countries worldwide by World Vision.[20] Additionally, the behaviour-change communication strategy was designed with the P Process tool, which uses narrative and negotiation to reach agreements with families to try new practices and has been used for more than 30 years for planning health-communication programmes.[21] The educational modules delivered during home visits were designed to address maternal, neonatal and child health, and included topics such as preparation of birth plan, facility-based birth, mother and newborn care, and folic acid importance (see online supplemental appendix for full list of educational modules). Sessions were also designed to include discussion of relevant regional practices that were identified during formative work in our pilot study as being potentially dangerous to newborns. Two such practices were the use of chupones during feeding (cloth materials dampened with saliva, honey or plant waters) and the placement of a fajero (abdominal cloth wrap) around a newborn's umbilical cord site.[17 18] Additional information regarding the planned protocol for each session and details of ttC method integration is available in online supplemental appendix and is additionally published elsewhere, including the supplementary material of our study protocol.[17 19]

CCAs used tablets during intervention delivery, which served as a visual and audio aid for educational materials (ie, families could see stories in video and listen to songs/riddles); as a prompt for CCAs to standardise intervention implementation; and as a medium to collect data.[22] Intervention delivery differed from other community-based approaches in that it was randomised to any household regardless of household composition (not just delivered to expectant mothers); it was comprised of long home visits and did not include additional outside messaging (ie, pamphlets, radio messages); and it did not include the use of community action groups. However, it should be noted that both intervention and control villages may also have received other interventions simultaneously, including both traditional behaviour change interventions through the Government of Honduras and supply-side interventions through IDB and the Salud Mesoamerica Initiative.[23]

We developed a structured survey instrument comprised both validated scales and internally developed items based on extensive literature review, expert consultation, input from the Ministry of Health of Honduras, and qualitative research with local residents and village leaders. Our survey instrument was primarily designed to capture target outcomes related to maternal, neonatal and child health, in addition to demographic information. On study enrolment, participants completed a photographic

census and baseline survey from June 2015 to June 2016, using our publicly available social network data collection software 'Trellis'.[24] An additional photographic census was conducted from January 2019 to December 2019, at which point participants also completed the endline survey. For more details on our pilot work and field operations see the published protocol.[17] No important changes to methods were made after trial commencement.

### Outcomes

The primary outcomes can be grouped into the following practices: (1) prenatal care for the mother, including folic acid use, creating a birth plan and receipt of prenatal care in first trimester; (2) giving birth in a health facility; (3) care-seeking for danger signs experienced by both mother and newborn; (4) postnatal care for mother, including preventative check-ups with health professionals; (5) postnatal care for newborn, including preventative check-ups with health professionals, proper thermal care and proper cord care (including withholding use of fajero); (6) immediate and exclusive breast feeding for infants under 6 months (including withholding use of chupones); (7) paternal involvement in pregnancy and child care; (8) proper treatment of diarrhoea and respiratory illness in children 5 and under; (9) reproductive health (including contraceptive use); and (10) gender/reproductive norms (see online supplemental appendix for full list of outcomes). Secondary outcomes were knowledge or attitudes related to many of the primary outcomes. All outcomes were measured via our endline survey instrument. The inclusion of certain responses to our survey questions was determined by eligibility encoded into the survey design, which was termed 'denominator' and can be found in online supplemental appendix table S1 for each outcome. For example, only mothers and fathers who had a child since the end of the intervention (1 September 2018) were eligible to be asked in the endline survey about whether their child was exclusively breast fed for the first 6 months.

### Statistical analysis

Given that our intervention was designed to be delivered at the household-level and that its behaviour change strategies are embedded in ideas of addressing community norms, we cannot rule out the presence of between-participant interference within villages. In particular, we assume that there could be interference within villages but not between villages (partial interference), and that a respondent's outcome may be affected by the proportion of treated households in the village in addition to their own treatment status (stratified interference).[25 26] This assumption is substantiated by an analysis in prior work in a different part of Honduras that showed enhanced adoption of a nutritional intervention consistent with social magnification.[27]

In this analysis, we use a two-stage randomisation design (in which we randomised village clusters to a dosage and then further randomised households to receive the intervention) in order to estimate: the total effect of the intervention on the treated (the effect of receiving the treatment in an intervention village compared with being in a control village); the direct effects on the treated (the effect of receiving the treatment versus not in an intervention village); and finally the spillover effects on the untreated who lived in the treated villages and may have been affected by others who received the intervention.[25] Here, we focus on the total effect, which should be interpreted as the effect of receiving the intervention in addition to being exposed to spillover effects from other targeted individuals in the village.

Under an intent-to-treat framework, we used multivariate logistic regression fitted to individual-level data for analysis of binary outcomes. We pooled responses across all treatment arms to obtain an average effect across all village-level treatment dosages, adding sampling weights to adjust for differential probability of treatment assignment. Clustered heteroskedasticity-robust standard errors that allowed for intragroup correlation were estimated at the level of villages, which was the unit of randomisation in the first stage. Further information about our analysis strategy can be found in online supplemental appendix. Significance was defined as $p < 0.05$. All analyses were performed using R V.4.0.4 and Stata V.13.

### Patient and public involvement

Community leaders of each of the villages were involved as partners in this study, and both the intervention team and the survey data collection team worked with local communities at each step of the study design process to ensure feasibility and incorporate feedback obtained through focus groups, interviews, pilot surveys and other outreach.

## RESULTS

Trial participants 12 and older were recruited and completed a baseline survey between June 2015 and June 2016. A total of 16 301 individuals across 5633 households were randomised, 13 881 of whom completed baseline surveys. And 10 263 individuals completed the endline survey between January 2019 and December 2019, approximately 24 months after intervention delivery began. Of 6038 individuals lost by the endline, 4% were lost to death, 50% to out-migration, 12% to refusal to continue in the study, 32% due to not being reached and 2% to other reasons. Those lost to follow-up were more likely to be younger, male, have a primary school or greater education, be single, not identify as Maya Chorti, and have greater self-rated physical and mental health, although notably there was no difference in attrition between households randomised to the intervention and those not (online supplemental appendix table S10).

A total of 77 villages with 4410 households were randomised to the random targeting strategy and 22 villages with 1223 households were randomised to the control arm with no treated households. Among the

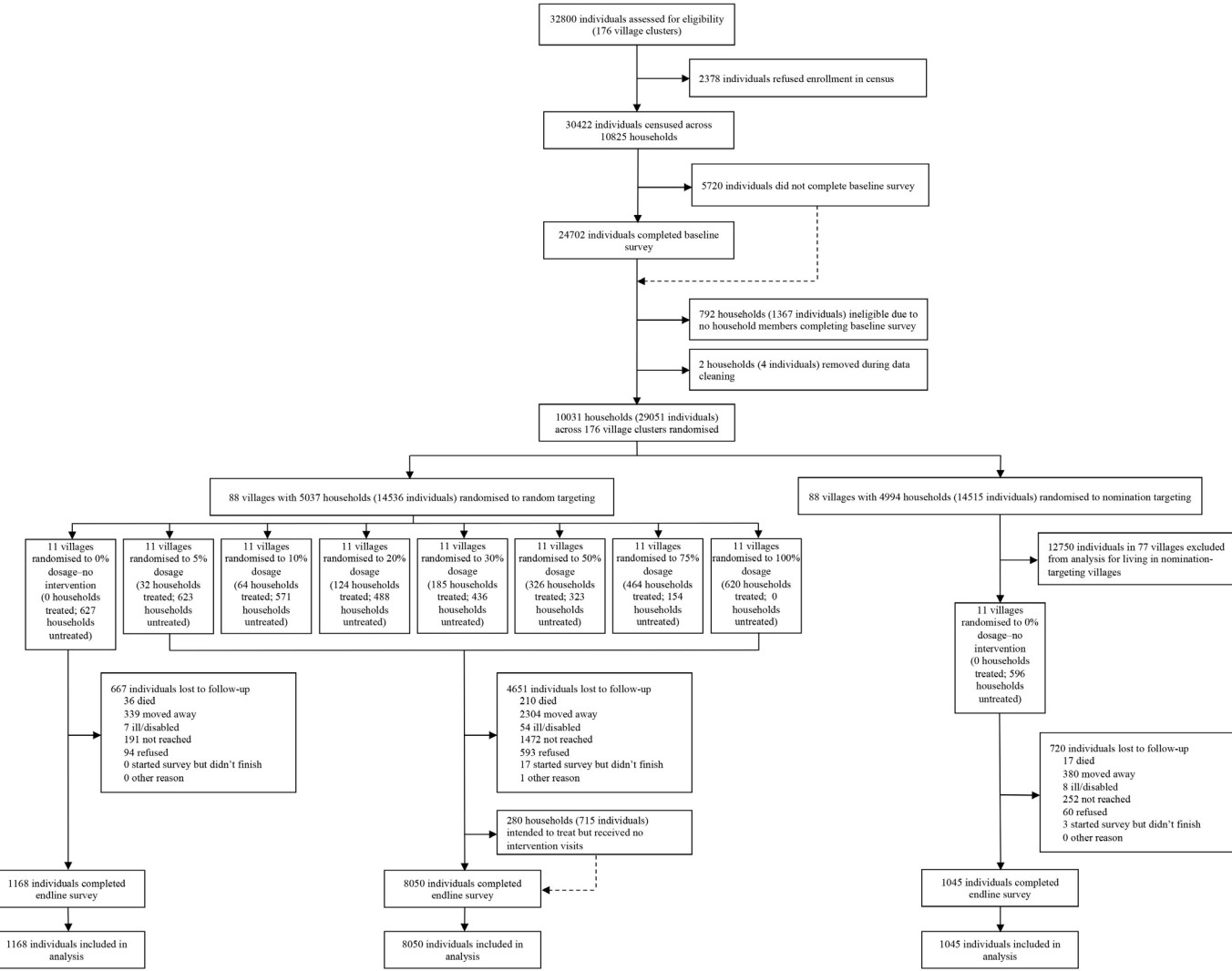

**Figure 1**  Trial profile: a population-level cluster-randomised controlled trial of maternal and child health intervention to promote behaviour change in Honduras.

77 villages assigned to the random targeting strategy, 11 villages were randomly assigned to each treatment dosage, or proportion of households targeted (figure 1, see online supplemental appendix figure S1 for parent trial profile). Across all treatment dosages, a total of 1815 households with 5305 individuals were randomised to receive the intervention. Villages had between 48 and 571 participants each, and households assigned to treatment received the intervention's educational household visits between November 2016 and August 2018.

At baseline, 8700 out of 16 301 (53%) participants were women. The mean age was 32.9 years (SD 17.2); 9404 (58%) respondents were married or in a civil union; and 8589 out of 13 881 (62%) participants had less than a primary education. Of the parents that completed an endline survey, 630 reported having a child after the intervention delivery period (1 September 2018) and 1133 reported having a child who was five or under at baseline. Descriptive statistics of the sample arranged by treatment arm are reported in table 1. Village and

household characteristics were balanced across treatment arms at baseline.

Significant total effects of the intervention on several primary and secondary outcomes are displayed in figure 2. Direct, indirect and total effects for each outcome are reported in online supplemental appendix table S1. We focus here on total effects of the intervention pooled across all dosages, which account for both the direct effect of the intervention and any spillover effects within each village (ie, the effect of having other treated individuals in the same village on a person who did not receive the intervention). Although all three effects are estimated, the total effects are emphasised as the best average estimate of the intervention's impact in the study population and as the metric most analogous to treatment effects reported from other comparable educational intervention trials.

Compared with parents of children born after the intervention period who did not receive the intervention, parents of children who received it showed significant

**Table 1** Baseline characteristics of the intention-to-treat population in the pure control and random-assignment arms

| | | Random-assignment group (n=12701) | |
|---|---|---|---|
| | **Pure control group (n=3600)** | **Within-cluster control (n=7396)** | **Targeted (n=5305)** |
| Household characteristics | | | |
| Wealth index | | | |
| Quintile 1 | 524 (15%) | 1194 (17%) | 896 (17%) |
| Quintile 2 | 558 (16%) | 1394 (19%) | 1073 (21%) |
| Quintile 3 | 592 (17%) | 1596 (22%) | 983 (19%) |
| Quintile 4 | 819 (22%) | 1487 (21%) | 1152 (22%) |
| Quintile 5 | 1080 (30%) | 1566 (21%) | 1116 (21%) |
| Households | 1223 | 2595 | 1815 |
| Household size* | 4.28 (2.1) | 4.35 (2.1) | 4.52 (2.2) |
| Child characteristics | | | |
| Youngest child's age, years* | 6.04 (5.2) | 5.44 (4.8) | 5.51 (4.9) |
| Past month illness (child 5 or under)* | | | |
| Respiratory illness | 281 (30%) | 674 (33%) | 527 (36%) |
| Diarrhoeal illness | 124 (13%) | 338 (16%) | 251 (17%) |
| Respondent characteristics | | | |
| Sex | | | |
| Female | 1908 (53%) | 3964 (54%) | 2828 (53%) |
| Male | 1692 (47%) | 3432 (46%) | 2477 (47%) |
| Age | 33.2 (17.4) | 32.8 (17.1) | 32.8 (17.2) |
| Education | | | |
| Less than primary | 1845 (60%) | 3907 (62%) | 2836 (63%) |
| Primary or greater | 1206 (40%) | 2438 (38%) | 1680 (37%) |
| Marital status | | | |
| Single | 1274 (35%) | 2672 (36%) | 1959 (37%) |
| Married or civil union | 2120 (59%) | 4254 (58%) | 3030 (57%) |
| Separated/divorced/widowed | 206 (6%) | 470 (6%) | 316 (6%) |
| Indigenous status* | | | |
| No | 2807 (92%) | 5489 (87%) | 3932 (87%) |
| Yes, Maya Chorti | 243 (8%) | 851 (13%) | 573 (13%) |
| Yes, other indigenous group | 1 (<1%) | 4 (<1%) | 2 (<1%) |
| Health* | | | |
| Self-rated physical health, 1 (excellent)–5 (poor)† | 3.21 (1.1) | 3.25 (1.1) | 3.24 (1.1) |
| Self-rated mental health, 1 (excellent)–5 (poor)† | 3.08 (1.1) | 3.12 (1.1) | 3.14 (1.1) |

Data are n, n (%) or mean (SD).
*Information only available from respondents who completed baseline survey (n=13881).
†Health questions from single item assessing general self-rated health (SF-1).[32]

improvements in primary outcome practices related to postnatal care, proper umbilical cord care and breast feeding. Specifically, the total effect of the intervention led to a 16.4% (95% CI 3.1%–29.8%, p=0.016) increase in the probability that parents had their newborn's health checked by a professional in a health facility within 3 days of birth; a 19.6% (95% CI 4.2%–35.1%, p=0.013) increase in the probability that parents did not wrap a fajero around the umbilical cord in the first week after birth; and a 8.9% (95% CI 0.3%–17.5%, p=0.043) increase in the probability that the mother breast fed immediately after birth (figure 2; online supplemental appendix table S1). The absolute change in behaviour for all targeted respondents and for all untargeted respondents (in both intervention and control villages) was on average: +13.9% for the targeted versus +5.8% for the untargeted for

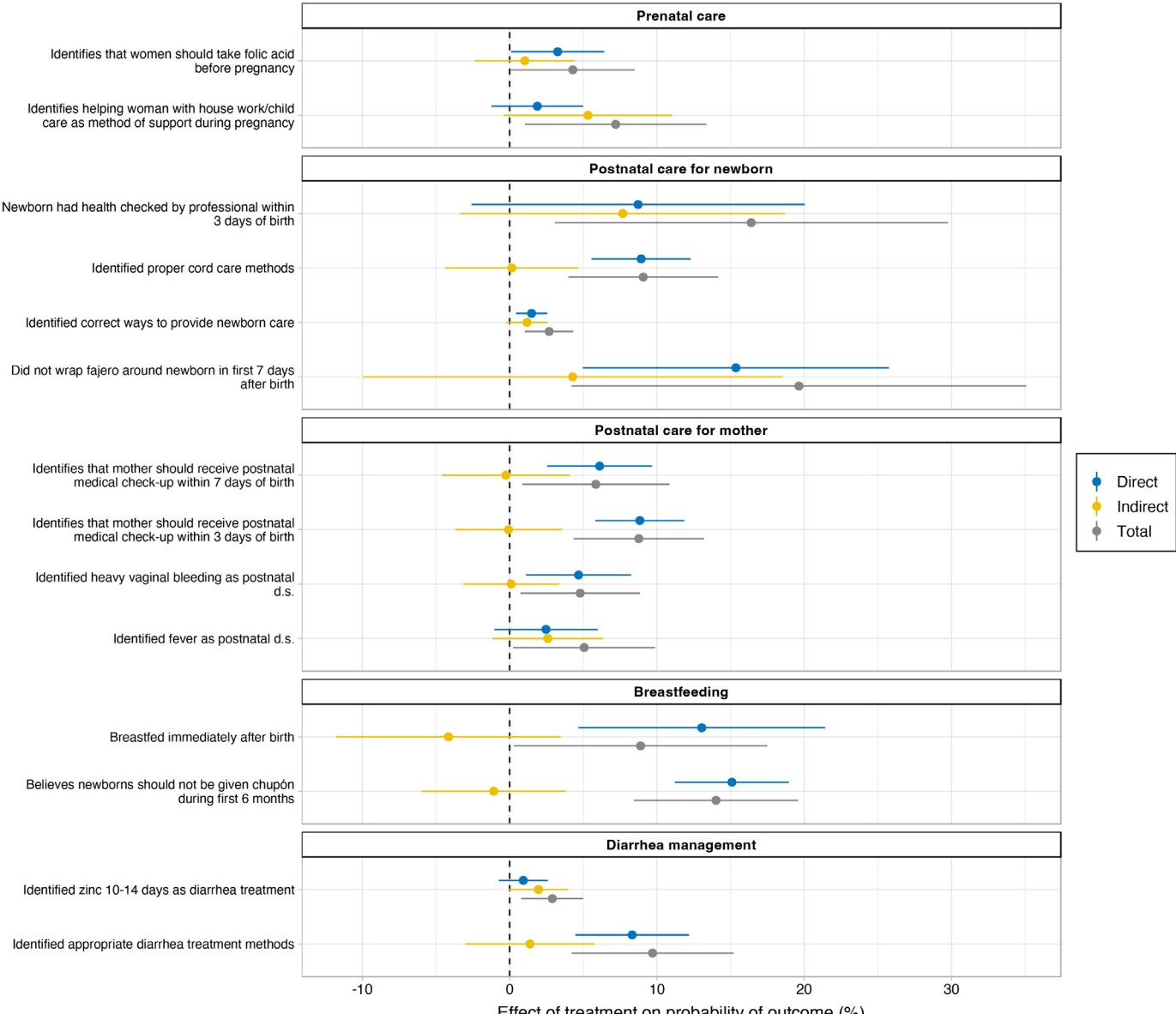

**Figure 2** Significant total, direct and indirect effects of the randomised controlled trial of maternal and child health intervention on prenatal/postnatal care, breast feeding and diarrhoea management.

having newborn's health checked within 3 days; +37.9% for the targeted versus +16.1% for the untargeted for not wrapping a fajero around the umbilical cord within the first week; and +5.1% for the targeted versus −4.1% for the untargeted for mothers breast feeding immediately after birth (online supplemental appendix table S9). Among parents of children born after the intervention period, we found no significant total effect of the intervention on practices related to preparing a birth plan, seeking care for certain pregnancy danger signs, facility based birth, mothers seeking postnatal check-up or care for certain postnatal danger signs, keeping the newborn wrapped or holding them skin-to-skin after birth, exclusively breast feeding for 6 months, or father involvement in newborn care. Additionally, we found no significant total effects on certain measures of diarrhoea or respiratory illness treatment in children five and under, folic

acid use among women 15 and over, or contraceptive use among all respondents.

In addition to the changes in actual behaviours related to maternal and child health, we also assessed changes in the knowledge and attitudes related to those behaviours as secondary outcomes. Changes in knowledge or attitudes may precede changes in practices, and we analysed responses of all participants regardless of whether they had a child since the intervention. We noted improvements in knowledge or attitudes related to the significant practices in the areas of postnatal care, proper umbilical cord care and breast feeding among the treated. Specifically, the total effect of the intervention was a 2.7% (95% CI 1.0%–4.3%, p=0.0014) increase in the probability that a participant identified correct ways to provide newborn care, which included getting their health checked by a medical professional; a 9.1% (95% CI 4.0%–14.2%,

p=0.0005) increase in the probability that a participant identified proper cord care methods, which included not wrapping a fajero; and a 14.0% (95% CI 8.4%–19.6%, p<0.0001) increase in the probability a participant believed newborns should not be given chupones when breast feeding in the first 6 months (figure 2; online supplemental appendix table S1).

Additionally, we found significant total effects for knowledge and attitudes related to folic acid use before pregnancy; methods of saving as part of the birth plan; certain pregnancy danger signs (ie, fever, swelling of hands/face/feet); fathers accompanying mothers to prenatal care visits; mothers receiving postnatal check-ups within 3 days of birth; certain postnatal danger signs for mother (ie, heavy vaginal bleeding, fever); identifying appropriate diarrhoea treatment methods including correct use of zinc; and delaying pregnancy until 18 years or older (figure 2; online supplemental table S1). We found no significant total effect of the intervention on knowledge or attitudes related to seeking prenatal care in the first trimester; facility-based birth; the majority of the pregnancy danger signs; the majority of the postnatal and newborn danger signs; immediate breast feeding; father waiting at birth location or caring for children when they are sick; diarrhoeal illness prevention; or respiratory illness prevention and danger signs (online supplemental appendix table S1).

No adverse events associated with the intervention were found. We performed additional analyses as a check for robustness of our results, which included looking exclusively at the response of the mothers, adjusting for demographic characteristics of respondents, adjusting for baseline behaviours and excluding those who moved between treatment arms during the study (online supplemental appendix tables S2–S7).

Finally, we incidentally note that two outcomes were associated with significant spillover effects on those who were untreated, which are effects of having others in one's village receive the intervention: father holding the child, and child with past-month diarrhoeal illness being treated with zinc (online supplemental appendix table S1). Our study design, which included variation in the fraction of people treated per village and the presence of a pure control group of wholly untreated villages allowed us to perform such an analysis. A comprehensive investigation of how spillover effects may depend on treatment dosage and effects specific to a village's social network structure are the subject of other work.

### Quality assessment of intervention delivery
Monitoring data on session implementation showed that the intervention was delivered as intended, with up to 22 monthly sessions planned per intervention village. Of the 1815 households initially targeted for intervention at baseline, 1225 (67%) households received modules for all 15 intervention topics as designed. Of the targeted households that completed endline surveys, the average number of sessions delivered per household was 15.2 (SD

7.4), with an average group size of 1.1 (SD 0.7) in attendance. Of the parents of children born after the intervention period who completed endline surveys, mothers were present for an average of 9.0 educational modules (SD 7.8) while fathers were present for an average of 2.4 educational modules (SD 2.8). Finally, as a method of assessing intervention delivery quality, we administered three survey questions designed as riddles for which the answer was based specifically on information covered in a session visit. We found significant (p<0.0001) effects of the intervention on answers to all quality-check questions (online supplemental appendix table S8).

### DISCUSSION
In this multi-arm cluster-RCT, we implemented an intervention targeting health outcomes for mothers, neonates and children which featured validated counselling and negotiation techniques delivered by CHWs to households in rural Honduras over the course of 22 months. Our results show improvements in many, but not all, practices related to postnatal care, proper umbilical cord care and breast feeding. Parents with children born after receiving the intervention were 16% more likely to report that their newborn's health was checked by a professional in a health facility within 3 days of birth; 20% more likely to report not using a fajero on the newborn's umbilical cord; and 9% more likely to report their child was breast fed immediately. Importantly, whenever we noted a significant effect on behaviour, we also saw an effect on the corresponding knowledge or attitude. We generally noted that our intervention seemed to have more impact on knowledge and attitudes than on practices, which is consistent with the theory that changes in knowledge and attitudes may be required first in order to seed changes in behaviour and that such changes are easier to affect.

Although our study design randomised villages to different dosages, or proportions of households treated, we estimate pooled effects across all dosages. The primary contribution of our findings, therefore, is that we provide estimates of average treatment effects. Such estimates represent the expectation policymakers can have on average in a setting where different proportions of the community are targeted for intervention (including observational settings in which the intervention is delivered non-randomly). Overall, however, the majority of pooled spillover effect estimates are negligible. Spillover effects may be heterogenous across dosage of households treated, social network structure or individual treatment status. Further investigation of spillover effects in this study remains the subject of other work.

The design and implementation of this trial had several strengths. The experimental design was robust and involved a two-stage randomisation in which we assigned village clusters to treatment dosages and then further assigned households within those clusters to treatment. Similar groups were attained through balancing across a number of village-level and household-level covariates

captured at the baseline stages of the project. In varying the dosage of households treated and randomising some villages to a pure control design in which no households were treated, we were able to estimate pooled spillover effects, a feature uncommon to other large-scale public health intervention evaluation. Careful monitoring and training of CCAs through partnerships with IDB resulted in unbiased, high-quality data acquisition across 4 years from baseline to endline. The selected behaviour change strategy was a modified version of the ttC methodology with the P Process tool, which together allowed CCAs to use dialogue counselling methods and interactive story-telling to assess current needs and practices of intervention recipients and negotiate on progressive improvement.[20 21] The approach has been documented as effective in changing a variety of behaviours aligned with the UN's Sustainable Development Goals and was successfully tailored to the conditions of the current study.[7 8 28] Finally, this study assessed a wider array of outcomes than other trials of its kind, which helped to determine the impact of the intervention on various points of the pregnancy and infancy timeline.

The study also had a number of limitations. In a traditional behaviour change communication intervention, households would be targeted based on where pregnant women or young children live, and community members would be exposed to the intervention messages as many times as possible (eg, in-person meetings, radio messages, flyers, etc). Neither of these were possible given our randomisation scheme (and the parent trial's larger goal of measuring spread of information and behaviour via social networks). Although we made significant attempts to capture a diversity of villages in our study, some were excluded on the basis of accessibility and safety. An additional consideration limiting generalisability was that respondents who were lost to follow-up were more likely to be younger, male, single, have a higher education, not identify with an indigenous group, and have greater self-ratings of physical and mental health. While we attempted to use objective measures whenever possible, both primary and secondary outcomes were assessed using self-report. Additionally, some practices targeted by our intervention were already performed at high rates in Honduras prior to our study, introducing ceiling effects to our intervention (eg, both DHS and our baseline survey show 86% of women began breast feeding immediately after birth; online supplemental appendix table S1).[13] Finally, our study area was subject to cointervention, the most notable of which was the Salud Mesoamérica Initiative (SMI). SMI involved both the Ministry of Health of Honduras and IDB and included implementation of both demand and supply-side interventions to improve maternal and child health within the Department of Copán.[23] All municipalities, and therefore villages, within Copán that were included in the current study were also included in SMI, and all were randomised to SMI's intervention arm. This suggests that both treated and untreated households in our trial experienced equal impact from SMI's

cointervention, and also that this trial's interaction with SMI within treated households may have slightly reduced the effect of our intervention. We took care to ensure that our intervention had unique and specific messaging that was evaluated through our outcome measures, and we note that the presence of cointervention improves our study's generalisability, as it is rare that large-scale public health interventions occur in isolation.

Many of our findings are consistent with similar home-based intervention packages targeting maternal, neonatal and child health through behaviour change communication strategies. Our review of the literature suggests that this is one of the largest RCTs of a maternal and child health intervention covering a full range of outcomes to take place in the Americas. Three studies reporting interventions across rural communities in Southern Ethiopia, India, Cambodia, Kenya and Zambia which used the ttC counselling strategy found a 10% improvement in immediate breast feeding, a 25% improvement in exclusive breast feeding within first 6 months, a doubling in family planning, and a 7.5-fold increase in skilled birth attendance.[7–9] One intervention also used Community Scorecards as a mechanism of social accountability, which was not used in the present study, and all used fewer home visits overall.[9] The evaluation of breast feeding in the current trial was largely based on the Alive & Thrive initiative, which relies on interpersonal communication and community mobilisation for home-visit counselling to improve breast feeding and complementary infant feeding practices in low-income regions.[10] Alive & Thrive's 2019 trial in rural Burkina Faso used messaging about feeding through both conversation and physical materials at health centre consultations, women's support groups, home visits, and public community events, and targeted pregnant or breast feeding women and their family members. Our findings of impacts on knowledge and attitudes about exclusive breast feeding are in agreement with their trial results, with their trial showing additional effects on exclusive breast feeding practices up to 39%.[10]

Even among trials like Alive & Thrive targeting similar outcomes with similar behaviour change communication strategies, almost no other study (to our knowledge) had exclusive messaging by CHWs during in-person home visits, nor a targeting strategy that selected households regardless of current or recently pregnant mother status. Two quasi-experimental interventions in rural Indian and Ugandan villages—the Integrated Nutrition and Health Programme (INHP) and MANIFEST intervention, respectively—which used home visits by CHWs to promote prenatal, delivery and newborn care practices, each reported significant improvements in outcomes related to prenatal check-ups, facility delivery, umbilical cord care, thermal care and breast feeding.[29 30] However, the INHP was designed as an non-governmental organisation facilitation of a government programme with the primary outcome being the socioeconomic equity of changes in the targeted behaviours, while the MANIFEST

intervention involved media communication and health-care capacity-building components.[29][30] Additionally, almost no interventions included discussion of regional practices that may have cultural significance but be potentially harmful to the mother or newborn. Our study suggested that inclusion of such practices in our communication strategy was effective in impacting knowledge, attitudes and practices related to fajero use, as well as knowledge and attitudes about the use of chupones.

While our randomisation scheme and large sample size increased our confidence that observed differences in outcomes, or lack thereof, are due to the intervention, further investigation into why change may or may not occur is warranted. For example, many of the primary outcomes were prevalent at baseline, including immediate breast feeding (86%), avoiding harmful substances around umbilical cords (83%) and keeping the newborn warm and clothed after birth (99%) (online supplemental appendix table S1). An even larger swath of secondary knowledge and attitudes outcomes were prevalent at the 80% level or above at baseline (online supplemental appendix table S1). High baseline rates of certain practices, knowledge and attitudes may limit the generalisability of the current study, and more work is needed to further clarify how a similar intervention may have a different impact in other communities, as well as how educational intervention packages in general might be tailored to account for the regional prevalence of behaviours which impact maternal and child health.

Additionally, the intervention was designed for behaviour change and was therefore aimed at the demand-side conditions of health behaviours. In many cases, however, said behaviours may not only rely on prior practices, knowledge and attitudes, but also on supply-side conditions such as the accessibility of health-care resources or the quality of local health centres. For example, outcomes evaluated in the current study but which depend on supply-side conditions include folic acid use, preventative check-ups, care-seeking from medical professionals due to the experience of either postnatal or newborn danger signs, delivery in a health facility and zinc treatment for diarrhoeal illness. Other outcomes were not as dependent on supply-side conditions and were therefore considered in the household's control and potentially more able to be impacted by a counselling intervention, such as preparing a birth plan, immediate and exclusive breast feeding, withholding chupones and fajeros, keeping the newborn warm and clothed, and skin-to-skin contact with newborn.

Overall, a novel large-scale household-based counselling intervention package randomised at the village and household levels and delivered in a rural setting resulted in meaningful change in practices, knowledge and attitudes related to preventative care-seeking for newborns, proper umbilical cord care and immediate breast feeding. Additional studies are needed to understand the underlying reasons for change in some but not all outcomes. In our opinion, the results of the trial are generalisable to similar communities in this region. Further research is also needed to assess the cost-effectiveness of this approach and its generalisability to other areas or practices. Finally, our two-stage randomisation allowed us to identify distinct treatment effects, suggesting that there are factors such as social network ties and community structure which can influence the uptake of behaviour change.[31] Further evaluation of the role of social network ties is needed to understand how, when people are given public health education, their response depends on how others around them are coping with similar challenges.

**Author affiliations**
[1]Yale Institute for Network Science, Yale University, New Haven, Connecticut, USA
[2]Institute for Health Equity Research, Icahn School of Medicine at Mount Sinai, New York, New York, USA
[3]Inter-American Development Bank, Washington, District of Columbia, USA
[4]Department of Statistics, Operations, and Data Science, Fox School of Business, Temple University, Philadelphia, Pennsylvania, USA
[5]Data Science Institute, Temple University, Philadelphia, Pennsylvania, USA
[6]Departments of Sociology and Medicine, Yale University, New Haven, Connecticut, USA
[7]Department of Biostatistics, Yale University, New Haven, Connecticut, USA

**Acknowledgements** We acknowledge our partners at the Bill and Melinda Gates Foundation and all staff of the Human Nature Lab at Yale University, including: HNL Lab and Project Director Thomas Keegan; Data Scientist Liza Nicoll; software developers Mark McKnight and Wyatt Israel; research coordinators Jai Broome, Catherine Henry, Kevin Garcia, Petergaye Murray, Samy Galvez, Maria Vassimon de Assis; as well as research assistants John Sang Won Lee, Selena Lee and Drew Prinster. We acknowledge our staff at IDB including Dr Hugo Godoy, IDB Health Specialist in Honduras, the Salud Mesoamerica Initiative Team, especially Maria Paola Zuniga, Mauricio Perez Calvo, Jose Ignacio Mata, and María Elena Ordóñez, IDB consultant, and partners from the World Vision Honduras Team including Karen Ramos, Lesbia Garcia, Gertrudis Medrano, and the Dimagi Team who supported the development of the community health application used during the intervention. We also acknowledge our partners at the Ministry of Health Honduras and all our local partners and field data collection team, and we thank all study participants.

**Contributors** MA, NAC, EMA, and LF formulated the hypothesis and conceptualised this study. World Vision Honduras directed the design of the intervention content, supervised by IDB, and the content was then reviewed by HNL to ensure that it met study criteria. NAC, RN, JN and EI were involved in survey design. RN trained data collection teams and managed study implementation, and JN and EI oversaw intervention delivery. NAC and RN managed data collection. WO, MA and LF analysed the data, and MA, WO, EMA, LF and NAC interpreted the analyses. WO did the literature search. WO, MA and LF produced the figures and tables, and, with RN and NAC, wrote the first draft. WO, LF and RN wrote the online supplemental appendix. All authors reviewed and provided input to the final draft. NAC and LF had final responsibility for the decision to submit for publication. LF is the guarantor of this work and, as such, had full access to all data in the study and takes responsibility for the integrity of the data and accuracy of the analysis.

**Funding** The trial was an investigator-initiated study supported by grants from the Bill & Melinda Gates Foundation and the Tata Group. Additional support was also provided, in part, by NSF awards CAREER IIS-1149662 and IIS-1409177, by ONR awards YIP N00014-14-1-0485 and N00014-17-1-2131, and by NIH award R01MH134715. The funding sources did not have any role in the design, conduct, or analysis of the study, the writing of the report, or the decision to submit it for publication. All authors had full access to all the data in the study. NAC and LF had final responsibility for the decision to submit for publication.

**Competing interests** NAC reports grants from the Bill and Melinda Gates Foundation and the Tata-Yale Alliance. EMA reports grants from the National Science Foundation and Office of Naval Research. LF reports a grant from the National Institutes of Health. The Bill and Melinda Gates Foundation funded the intervention through a grant to IDB supervised by JN and EI. All other authors declare no competing interests.

**Patient and public involvement** Patients and/or the public were involved in the design, or conduct, or reporting, or dissemination plans of this research. Refer to the Methods section for further details.

**Patient consent for publication** Not applicable.

**Ethics approval** This study involves human participants and all participants provided informed consent at the time of data collection before randomisation. Ethics approval was obtained from the Institutional Review Board (Protocol #1506016012) at Yale University in New Haven, Connecticut, USA and the Ministry of Health of Honduras.

**Provenance and peer review** Not commissioned; externally peer reviewed.

**Data availability statement** Data are available upon reasonable request. The de-identified datasets generated during the study along with the statistical plan and analytic code will be available from the corresponding author on reasonable request at the end of the project after all planned manuscripts have been accepted for publication. We will make the data without identifiers available to users only under a data-sharing agreement that provides for: (1) a commitment to using the data only for research purposes and not to make the attempt of identifying any individual participant; (2) a commitment to securing the data in case there are still some sensitive variables after the identifiers have been removed, by using appropriate computer technology; (3) a commitment to destroying or returning the data after analyses are complete; and (4) a commitment to not publish any information that is not treated at the aggregate level so that no specific characteristics can be linked to small communities.

**Open access** This is an open access article distributed in accordance with the Creative Commons Attribution 4.0 Unported (CC BY 4.0) license, which permits others to copy, redistribute, remix, transform and build upon this work for any purpose, provided the original work is properly cited, a link to the licence is given, and indication of whether changes were made. See: https://creativecommons.org/licenses/by/4.0/.

**ORCID iDs**
William Oles http://orcid.org/0000-0003-1125-6717
Marcus Alexander http://orcid.org/0000-0001-8627-7534
Nicholas A. Christakis http://orcid.org/0000-0001-5547-1086
Laura Forastiere http://orcid.org/0000-0003-3721-9826

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
