## [Reviewer comments · BMJ Open]

ARTICLE DETAILS

TITLE (PROVISIONAL)	Maternal and child health intervention to promote behavior change: a population-level cluster-randomized controlled trial in Honduras
AUTHORS	Oles, William; Alexander, Marcus; Negron, Rennie; Nelson, Jennifer; Iriarte, Emma; Airoidi, E; Christakis, Nicholas; Forastiere, Laura

VERSION 1 – REVIEW

REVIEWER	Winch, Peter Johns Hopkins University Bloomberg School of Public Health
REVIEW RETURNED	27-Aug-2022

GENERAL COMMENTS	1. This is a very well organized and well documented study. I had many questions while reading the manuscript concerning methods and study design. However, these questions were all answered in the study protocol paper Shakya et al. BMJ Open 2017. It would be helpful to the reader to be more explicit about the existence of this published paper, and direct the reader to it, at the beginning of the Methods section.2. Might also provide estimates for Honduras for Total Fertility Rate, age of marriage, Maternal Mortality Ratio, female literacy around lines 26 to 29. See for example https://dhsprogram.com/Countries/Country-Main.cfm?ctry_id=142. Need to know female literacy to assess appropriateness of counseling materials, and need for materials adapted for illiterate populations. It is stated in the Introduction that Honduras lags behind other LMIC in Latin America. Should compare under-five mortality rate and neonatal mortality rate to Mexico, and neighbors of Honduras: Guatemala, Nicaragua, El Salvador based on DHS surveys https://dhsprogram.com/publications/publication-FR274-DHS-Final-Reports.cfm or other comparable data.3. The intervention content is not described almost at all in the main body of the paper. Even in the Online Supplementary Appendix, the reader does not get a sense for how the intervention was delivered. How long did a counseling session / household visit last? What time of day did it occur? What visual aids supported the counseling? It would be valuable to have the entire protocol/ plan for one of the counseling sessions.4. What other interventions were in place in the villages prior to this study? What interventions were World Vision Honduras and Dimagi already implementing? To what extent was this intervention building on already-established infrastructure for delivery of interventions established by the NGOs?5. Analysis is solid, and very good statement of limitations. Another limitation that might be mentioned is that Honduras already has
---

	high values for many indicators such as Married women currently using any modern method of contraception, Demand for family planning satisfied by modern methods, and Place of delivery: Health facility, see https://dhsprogram.com/Countries/Country-Main.cfm?ctry_id=142 . Hence, Honduras did not have much room to improve, although in this remote rural area undoubtedly there was and still is considerable room to improve. The same intervention package, delivered in sub-Saharan Africa, might demonstrate much stronger effects, with indicators starting at a much lower baseline.
--	--

REVIEWER	Hoffman Goedert, Martha Seattle University - College of Nursing
REVIEW RETURNED	20-Dec-2022

GENERAL COMMENTS	This manuscript has been a pleasure to review. The impact of using basic health education, community health workers, creating gender affirming actions for both mothers and fathers, encompassing the global work from other country's maternal child life-saving work is exemplary. This can be used to guide replication of work in other global south countries, and to impress upon other global north countries, the importance of very basic, health affirming measures. I congratulate you. You have built upon the experience of other NGO's and international work and have created an impact that will stand.
---

REVIEWER	Nash, Denis City University of New York System, Epidemiology and Biostatistics
REVIEW RETURNED	09-Feb-2023

GENERAL COMMENTS	This is a well-written manuscript describing a large cluster RCT to improve pregnancy outcomes and infant outcomes via intensive educational interventions to change behaviors. The findings suggest that the intervention worked across a number of key outcomes. The innovative design of the trial allowed the authors to assess spillover effects of the intervention to those in the community who did not receive the intervention directly (ie, were in a community that was randomized, but not in a household that was randomized to receive the intervention). The team is to be congratulated for completing such an extensive investigation. However, there are also some limitations, nearly all mentioned by the authors, that may have undermined efforts to assess the true impact of the intervention, including the exclusive use of self-reported primary and secondary outcomes, the large number of primary and secondary outcomes with no correction for multiple comparisons, and a focus on relative changes from baseline as opposed to absolute change to assess the impact of the intervention. Specific comments: 1. The authors should include details on the absolute changes from baseline in their abstract and results section for both intervention and control groups. The Figure only appears to present the relative change. Related, the authors mention that the communities were already at fairly high levels for many outcomes, raising the possibility of a ceiling effect. It also speaks to
---

	generalizability, since it may be that areas with worse outcomes could see more benefit than was seen in this trial. 2. The authors refer readers to the published protocol (reference 17), but it is not clear where it is published and there is no link or DOI number to be able to find it. Without one, I would suggest including this as an appendix in their final article. 3. The authors mention that there were other related interventions happening simultaneously across intervention and control communities and households. This is always a risk with long term, real world studies, and I believe can actually help with generalizability. However, it would have been good to know more about the timing of these other initiatives, and the extent to which there may have been differential reach of these initiatives by study arm (and if so which study arms may have been more or less affected). This is important context for any cluster RCT, in my view, and should be provided/described if any information is available. 4. I could not tell if the control households or communities were attention matched in any way. clearly they all received the same interviews for baseline and follow-up/outcome measurements. Please report on the number of encounters by arm and the mean length of time for each encounter. But did interviewers develop the same rapport with households in control communities? This speaks to the extent to which study arms were similar in as many ways as possible except for the intervention. Clearly this is a challenge to do in uncontrolled, real world settings at scale. So to what extent could it have contributed to the observed findings. For example, might intervention households be more subject to social desirability bias. (related to this, have the authors considered using interview identity as a potential instrumental variable to assess and possibly correct for such bias? There are examples of using Hickman selection models in the HIV literature that may be helpful for this. see: McGovern ME, Barnighausen T, Salomon JA, Canning D. Using interviewer random effects to remove selection bias from HIV prevalence estimates. BMC Med Res Methodol. 2015;15:8.) 5. The sample sizes of some of the key sub groups(e.g., those who gave birth after the intervention) were not apparent. 6. There were many primary and secondary outcomes. As a result, there was not enough detail on the outcome definitions. For example, how soon after a birth was the outcome assessed on average? How did this differ by study arm? 7. Related, and perhaps more importantly is that the analyses should probably have corrected for the multiple comparisons that were made. For this, I would defer to a trial statistician. But the idea is that if you make enough comparisons, there is a higher chance of finding things as significant. It was reassuring that some of those things that were significant were accompanied by changes in other variables along the pathway from the intervention to the outcome. But this is a statistical inference issue and best practices should be followed.
--	---

REVIEWER	Thomas, Roger University of Calgary, Family Medicine
-----------------	---

GENERAL COMMENTS

This is a major achievement to conduct such a complex intervention over 22 months in a remote area and in Spanish. Congratulations!

Abstract:

Three statistically significant outcomes are presented but seven with no significant results are not presented (“cherry picking”).
[Please correct]

Purpose:

A 22-month household-based counseling intervention in Honduras was designed to improve practices, knowledge, and attitudes about maternal, neonatal, and child health.

Sample:

The eligibles were residents in 238 villages in one area, of which 176 were selected on the basis of “population size, geographic diversity, accessibility, and safety.”

The effect of this selection on generalisability was not assessed by analysing differences between the selected and not selected villages.

[Please address generalisability issues]

The final sample comprised 16,301 people in 5,633 households in 99 villages in the Copán region of Honduras

In each village a proportion of households (0, or 0.05, 0.1, 0.2, 0.3, 0.5, 0.75, 1) were randomly assigned to the intervention. Then households were randomly assigned to the intervention according to the village’s dosage with covariate-constrained randomization to balance treatment proportions at the village and household levels. This strategy was effective as shown in Table 1 with baseline variables very similar between the Pure Control, the Random assignment within cluster control and the Targeted intervention groups.

Outcome measures:

The survey instrument included validated scales and internally developed items based on literature review, expert consult, Ministry of Health of Honduras advice and qualitative research with local residents and village leaders.

A detailed analysis of the validity and reliability of the survey instruments was not provided.

[please correct]

Analysis

The analysis was intention-to-treat. Individual-level data were analysed by multivariate logistic regression with binary outcomes. How much data was lost by making outcomes binary was not stated.

One goal was to assess the average effect across all village-level treatment dosages and responses were pooled across all treatment arms and sampling weights were added to adjust for differential probabilities of treatment assignment. Clustered

	heteroskedasticity-robust standard errors that allowed for intragroup correlation were estimated at the village level. Results: Of the sample of 16,301, 13,881 completed baseline surveys and 10,263 completed the final survey 24 months later. Of the 6,038 individuals lost to the final survey, 50% migrated, 32% could not be reached, 12% refused to continue the study, and 4% died. There was no analysis of differences between completers and non-completers and effects on generalisability. [please comment on generalisability] Of the total 16,301 individuals across 5,633 households who were randomized, 77 villages with 4,410 households were randomized to the random targeting strategy and 22 villages with 1,223 households were randomized to the control arm with no treated households. Among the 77 villages assigned to the random targeting strategy, 11 villages were randomly assigned to each treatment dosage, or proportion of households targeted. Across all treatment dosages, a total of 1,815 households with 5,305 individuals were randomized to receive the intervention. Participant numbers varied widely from 48 to 571 between villages. Three outcomes had statistically significant outcomes, with very wide 95% CIs. There was a 16.4% (95% CI 3.1%–29.8%, $p=0.016$) increase in the probability that parents had their newborn's health checked by a professional in a health facility within three days of birth; a 19.6% (95% CI 4.2%–35.1%, $p=0.013$) increase in the probability that parents did not wrap a fajero around the umbilical cord in the first week after birth; and an 8.9% (95% CI 0.3%– 17.5%, $p=0.043$) increase in the probability that the mother breastfed immediately after birth. [please comment on the reasons for the very wide 95%CIs and implications for the ability to generalise the results of the implementation of your study] However, there were no significant effects of the intervention for seven other outcomes: knowledge or attitudes related to seeking first trimester prenatal care, facility-based birth; majority of the pregnancy danger signs; majority of the postnatal and newborn danger signs; immediate breastfeeding; father waiting at birth location or caring for children when infants were sick; prevention of diarrheal illness; or respiratory illness prevention and danger signs. No explanations for the non-significance of these outcomes were discussed. [please discuss these hypotheses which were not supported and reasons why not]
--	---

VERSION 1 – AUTHOR RESPONSE

Reviewer: 1

Dr. Peter Winch, Johns Hopkins University Bloomberg School of Public Health

Comments to the Author:

1. This is a very well organized and well documented study. I had many questions while reading the manuscript concerning methods and study design. However, these questions were all answered in the study protocol paper Shakya et al. BMJ Open 2017. It would be helpful to the reader to be more explicit about the existence of this published paper, and direct the reader to it, at the beginning of the Methods section.

Response: We thank the reviewer for their careful reading of our manuscript. The first paragraph of the “Methods” section has been amended to more explicitly direct the reader to our published protocol.

2. Might also provide estimates for Honduras for Total Fertility Rate, age of marriage, Maternal Mortality Ratio, female literacy around lines 26 to 29. See for example https://dhsprogram.com/Countries/Country-Main.cfm?ctry_id=142. Need to know female literacy to assess appropriateness of counseling materials, and need for materials adapted for illiterate populations.

Response: We agree that adding the above estimates provides useful context for the counseling intervention on which we are reporting. We have added text to this end in the third paragraph of the Introduction around the above-mentioned lines using data from the Honduras Demographic and Health Survey (DHS) dataset as suggested. Our local health education team had extensive training and experience in delivering the validated materials.

It is stated in the Introduction that Honduras lags behind other LMIC in Latin America. Should compare under-five mortality rate and neonatal mortality rate to Mexico, and neighbors of Honduras: Guatemala, Nicaragua, El Salvador based on DHS surveys <https://dhsprogram.com/publications/publication-FR274-DHS-Final-Reports.cfm> or other comparable data.

Response: We agree that it is helpful to provide evidence for how Honduras compares to other LMIC in Latin America and have added text to this end to the third paragraph of the Introduction, using the same dataset that we originally cited for Honduras’ child mortality statistics (collected by the UN Inter-agency Group for Child Mortality Estimation).

3. The intervention content is not described almost at all in the main body of the paper. Even in the Online Supplementary Appendix, the reader does not get a sense for how the intervention was delivered. How long did a counseling session / household visit last? What time of day did it occur? What visual aids supported the counseling? It would be valuable to have the entire protocol/ plan for one of the counseling sessions.

Response: General information about intervention content is provided within the “Methods - Procedures” section. Because the content of each home visit within the intervention varied, by design, in both content (across the 15 educational modules) and delivery (as part of the flexibility within the timed and targeted counseling methodology), we felt that it was beyond the scope of the manuscript body to provide more granular detail about intervention content.

However, in response to this comment, we have added additional details about the length and protocol of each household visit in the Supplementary Appendix within the “Intervention sessions” section. We have also added text to the manuscript body within the “Methods - Procedures” section to advertise this appendix addition and to reference a publicly-available publication of the home visit design and protocol as part of “Project REDES.” Our quality assessment of the success of intervention delivery is described within the “Results - Quality assessment of intervention delivery” section. We hope these additions are sufficient to clarify and reference the intervention content.

4. What other interventions were in place in the villages prior to this study? What interventions were World Vision Honduras and Dimagi already implementing? To what extent was this intervention building on already-established infrastructure for delivery of interventions established by the NGOs?

Response: These questions are incredibly important in the consideration of the impact of the intervention we describe in the current study. It is not possible to describe and quantify all public and private health interventions that were implemented concurrently to the current study across the same time period. However, the most well-published and likely significant intervention related to ours, which was also implemented by our partner Inter-American Development Bank, was the Salud Mesoamerica Initiative (SMI).

SMI is a multi-country, multi-phase public-private collaboration that aimed to improve maternal and child health conditions in the poorest populations of Mesoamerica, including Honduras. Interventions were all evidence-based and included implementation of the Essential Obstetric and Neonatal Care (EONC) strategy, strengthening referral networks, improving the supply chain, encouraging the cultural adaptation of services for indigenous populations, supporting new service delivery platforms and community platforms, and the design and approval of updated country norms and protocols (Mokdad et al., 2018). These interventions were carried out from 2015-2017, overlapping with the implementation of our project.

On our review, the 4 municipalities (Cabañas, Copán Ruinas, San Jerónimo, Santa Rita) containing the 176 villages randomized in our parent trial were all included in SMI, and all 4 were randomized to SMI’s intervention arm. There were an additional 4 municipalities which were not included in our trial but that are located in Copán and were included in SMI’s control arm. Therefore, any effects of co-intervention by SMI should have been experienced equally among all villages and households included in our trial. Additionally, although there were likely concurrent public health interventions, including SMI, ongoing at the time of our study, our criteria for village randomization included consideration of indicators of village wealth and health care access (supplemental appendix, “Blocking of villages and randomization”), and so other interventions should not have a significant effect on our effect estimation.

Finally, it is worth noting that SMI was an intervention implemented at the municipality level, while our intervention was implemented on a household level. Therefore, our estimation of the total and direct effects (which compare targeted and untargeted households within villages assigned to control or to the intervention, respectively) is a conservative one, given that untargeted households in our trial would have potentially “benefited” from the SMI intervention, while the interaction of our intervention with that of SMI on targeted households may have reduced the effects of our intervention.

In response, we have added text to highlight discussion of co-intervention effects when we expand on our study’s limitations in the fourth paragraph of the “Discussion” section.

Mokdad AH, Palmisano EB, Zúñiga-Brenes P, et al. Supply-side interventions to improve health: Findings from the Salud Mesoamérica Initiative. *PLoS One* 2018;13(4):e0195292. doi: 10.1371/journal.pone.0195292.

5. Analysis is solid, and very good statement of limitations. Another limitation that might be mentioned is that Honduras already has high values for many indicators such as Married women currently using any modern method of contraception, Demand for family planning satisfied by modern methods, and Place of delivery: Health facility, see https://dhsprogram.com/Countries/Country-Main.cfm?ctry_id=142. Hence, Honduras did not have much room to improve, although in this remote rural area undoubtedly there was and still is considerable room to improve. The same intervention package, delivered in sub-Saharan Africa, might demonstrate much stronger effects, with indicators starting at a much lower baseline.

Response: We agree that high baseline rates of certain health-related practices in Honduras is a significant limitation of the current study and introduces a ceiling effect to our evaluation of the intervention. We have added text to this end to our discussion of our study's limitations in the fourth paragraph of the "Discussion" section, highlighting an example of high baseline rates of immediate breastfeeding citing the above-mentioned DHS dataset.

Reviewer: 2

Martha Hoffman Goedert, Seattle University - College of Nursing

Comments to the Author:

This manuscript has been a pleasure to review. The impact of using basic health education, community health workers, creating gender affirming actions for both mothers and fathers, encompassing the global work from other country's maternal child life-saving work is exemplary. This can be used to guide replication of work in other global south countries, and to impress upon other global north countries, the importance of very basic, health affirming measures. I congratulate you. You have built upon the experience of other NGO's and international work and have created an impact that will stand.

Response: We thank this reviewer very much for their expression of support for our study and join them in the hope that we can contribute to the international mission of making progress towards better health outcomes for mothers and children through community engagement and emphasis on health-affirming measures.

Reviewer: 3

Dr. Denis Nash, City University of New York System

Comments to the Author:

This is a well-written manuscript describing a large cluster RCT to improve pregnancy outcomes and infant outcomes via intensive educational interventions to change behaviors. The findings suggest that the intervention worked across a number of key outcomes. The innovative design of the trial allowed the authors to assess spillover effects of the intervention to those in the community who did not receive the intervention directly (ie, were in a community that was randomized, but not in a household that was randomized to receive the intervention). The team is to be congratulated for completing such an extensive investigation. However, there are also some limitations, nearly all mentioned by the authors, that may have undermined efforts to assess the true impact of the intervention, including the exclusive use of self-reported primary and secondary outcomes, the large number of primary and secondary outcomes with no correction for multiple comparisons, and a focus on relative changes from baseline as opposed to absolute change to assess the impact of the intervention.

Response: We would like to thank the reviewer for their careful reading of our manuscript and thoughtful reflection on the strengths and weaknesses of the study. The limitations mentioned above regarding the correction for multiple comparisons and focus on relative versus absolute changes are addressed under specific comments below.

With respect to the use of self-reported primary and secondary outcomes, we agree that this was a limitation of the outcome design and list it both in the bulleted strengths and limitations section at the beginning of the manuscript and in the fourth paragraph of the “Discussion” section. The reasons for choosing self-reported outcomes are multifold. The nature of the timed and targeted counseling methodology within our intervention package allowed families to interact with survey instruments in real-time, both before and after each counseling session, such that their responses most closely reflected the quality and impact of the education-based intervention. Additionally, due to the large scale of the study, both in respondent number and geographic area, self-reported survey responses were the most logistically feasible way to collect data about a wide range of maternal and child health indicators at an individual and household level. Finally, because the parent trial of the current study was designed to investigate the effects of community network structure on intervention impact, and because dissemination of knowledge and attitudes as precedents to changes in behavior lies at the center of the social theory behind this design, we felt that it was important to include outcomes which assessed subject views about knowledge, attitudes, and behaviors related to intervention content.

For more details on the issue of multiple comparisons, see Reviewer 3 (comment 7) below.

Specific comments:

1. The authors should include details on the absolute changes from baseline in their abstract and results section for both intervention and control groups. The Figure only appears to present the relative change.

Response: We agree that a report of absolute changes in primary and secondary outcomes would add useful context to our pooled effects analysis. We have added an additional table, Table S9, to our supplementary appendix reporting absolute changes from baseline to endline stratified by target status with t-tests for significant differences between groups. We have also added these results to the abstract and “Results” section of the manuscript where appropriate.

Related, the authors mention that the communities were already at fairly high levels for many outcomes, raising the possibility of a ceiling effect. It also speaks to generalizability, since it may be that areas with worse outcomes could see more benefit than was seen in this trial.

Response: We agree that high baseline rates of certain behaviors may have introduced a ceiling effect to our estimation of the intervention’s impact. A similar comment was raised by Reviewer 1 (comment 5). We have added text to our “Discussion” section in our review of the study’s limitations to emphasize this point and its implications for our study’s generalizability.

2. The authors refer readers to the published protocol (reference 17), but it is not clear where it is published and there is no link or DOI number to be able to find it. Without one, I would suggest including this as an appendix in their final article.

Response: We have re-emphasized our published protocol in the revised manuscript in response to Reviewer 1 (comment 1), and we have additionally ensured that our citation of the study protocol includes a DOI for easier reference.

3. The authors mention that there were other related interventions happening simultaneously across intervention and control communities and households. This is always a risk with long term, real world studies, and I believe can actually help with generalizability. However, it would have been good to know more about the timing of these other initiatives, and the extent to which there may have been differential reach of these initiatives by study arm (and if so which study arms may have been more or less affected). This is important context for any cluster RCT, in my view, and should be

provided/described if any information is available.

Response: The issue of the effect of co-intervention on our ability to assess our intervention's validity is an important one and was additionally raised by Reviewer 1 (comment 4). We refer to our response there. The most notable concurrent intervention overlapping in time with our study was the second phase of the Salud Mesoamérica Initiative (SMI), which occurred from 2015-2017 and included all of the villages in the Department of Copán that were also randomized in this study's parent trial. On our review, the 4 municipalities (Cabañas, Copán Ruinas, San Jerónimo, Santa Rita) containing the 176 villages randomized in our parent trial were all included in SMI, and all 4 were randomized to SMI's intervention arm. We have added clarifying text in the fourth paragraph of the "Discussion" section to emphasize this point and provide reference to the SMI protocol.

4. I could not tell if the control households or communities were attention matched in any way. clearly they all received the same interviews for baseline and follow-up/outcome measurements. Please report on the number of encounters by arm and the mean length of time for each encounter. But did interviewers develop the same rapport with households in control communities? This speaks to the extent to which study arms were similar in as many ways as possible except for the intervention. Clearly this is a challenge to do in uncontrolled, real world settings at scale. So to what extent could it have contributed to the observed findings. For example, might intervention households be more subject to social desirability bias. (related to this, have the authors considered using interview identity as a potential instrumental variable to assess and possibly correct for such bias? There are examples of using Hickman selection models in the HIV literature that may be helpful for this. see: McGovern ME, Barnighausen T, Salomon JA, Canning D. Using interviewer random effects to remove selection bias from HIV prevalence estimates. *BMC Med Res Methodol.* 2015;15:8.).

Response: Our intervention design focused specifically on ensuring that any potential for surveyor-based bias or social desirability bias was minimized while inter-surveyor reliability was a priority. This was verified by on-going analyses of the survey responses after each reporting wave. Additionally, the team of interviewers who collected survey responses were not the same as the team that delivered the intervention, and therefore, there was no theoretical reason for which interviewers may have developed a different rapport with targeted households. Text clarifying this point has been added to the "Methods - Procedures" section.

We appreciate the idea of using random effects for interviewers or a Hickman selection model. However, given the length and breadth of this first publication of the results, we feel that this additional analysis, while potentially valuable, is beyond the scope of the current paper. Importantly, none of our current statistical analyses suggest that such surveyor random effects would have a substantive impact on our reported findings. We hope this is acceptable.

5. The sample sizes of some of the key sub groups (e.g., those who gave birth after the intervention) were not apparent.

Response: The primary subgroups that were pertinent to the reporting of our primary and secondary outcomes were respondents who were in randomized households and completed an endline survey (n=10,263) and respondents who reported having a child after the intervention delivery period ending September 1, 2018 (n=630). These numbers are reported in the first and second paragraphs of the "Results" section, respectively. We additionally direct readers to Table S1 in our supplementary appendix, where all pooled effects estimates are reported in addition to the subgroup size for each outcome. In particular, we reported both the "denominator" (the subgroup of respondents who were eligible to provide a survey response related to a specific outcome) and the actual subgroup size "N" (number of actual responses that were recorded and included in analysis for a given outcome).

6. There were many primary and secondary outcomes. As a result, there was not enough detail on the outcome definitions. For example, how soon after a birth was the outcome assessed on average? How did this differ by study arm?

Response: Our outcomes are defined in greater detail in the published protocol and the additional documentation provided in our supplementary appendix. There is no difference in the study arms in how these outcomes are defined with respect to the timing of perinatal or maternal care.

7. Related, and perhaps more importantly is that the analyses should probably have corrected for the multiple comparisons that were made. For this, I would defer to a trial statistician. But the idea is that if you make enough comparisons, there is a higher chance of finding things as significant. It was reassuring that some of those things that were significant were accompanied by changes in other variables along the pathway from the intervention to the outcome. But this is a statistical inference issue and best practices should be followed.

Response: All statistical analysis reported in the manuscript was designed by our study's trial statistician (LF). Importantly, as described in our trial's preregistration, we set out to report effects of our intervention for each of our primary and secondary outcomes independently of one another. It is absolutely true that if an investigator is interested in the impact of an intervention and wants to claim that the intervention is effective if any of the effects on multiple outcomes, pre-registered or not, are statistically significant, then the analysis would require adjustment for multiple testing. This is because this claim involves a composite null hypothesis that none of the outcomes are affected by the intervention vs the alternative disjunctive hypothesis that the intervention has an effect on at least one outcome, and this hypothesis testing does suffer from Type I error inflation (if the intervention was not effective, the chance of a spurious conclusion that it is effective because at least one effect was significant at the 0.05 level is much larger than 5%). However, here we do not test a single null hypothesis stating that the intervention would not have an overall effect on any of the outcomes. Rather, we tested many independent hypotheses, each stating that the intervention would not have an effect on a single, specific outcome that was of interest on its own. This design was intentional due to the unique nature of this study and its parent trial. Our survey instruments were lengthy enough to measure a broad array of health phenotypes, and the intervention we are evaluating contained fifteen educational modules, each with a different focus. Our analysis, therefore, did not require the use of adjustment for multiple comparisons (e.g., Rothman, 1990; Savitz DA, Olshan AF, 1995; Perneger, 1998). Importantly, because we made this choice in analysis design, we have tried to meticulously avoid making or suggesting the claim that the intervention is effective because a few of our outcomes are significant—as stated above, analysis of such a claim would rely on adjustment for multiple comparisons.

Rothman, KJ. No adjustments are needed for multiple comparisons. *Epidemiology* 1990;1(1):43–46. doi: 10.1097/00001648-199001000-00010

Savitz DA, Olshan AF. Multiple comparisons and related issues in the interpretation of epidemiologic data. *American Journal of Epidemiology* 1995;142(9):904-8. doi: 10.1093/oxfordjournals.aje.a117737

Perneger TV. What's wrong with Bonferroni adjustments. *BMJ* 1998;316(7139):1236-8. doi: 10.1136/bmj.316.7139.1236

Reviewer: 4

Prof. Roger Thomas, University of Calgary

Comments to the Author:

This is a major achievement to conduct such a complex intervention over 22 months in a remote area

and in Spanish. Congratulations!

Response: We thank this reviewer for their consideration of our study and their careful review of each section! The issues raised by the reviewer were very important and led to significant additions in both analysis and text to the revised manuscript.

Abstract:

Three statistically significant outcomes are presented but seven with no significant results are not presented (“cherry picking”).

[Please correct]

Response: We have added text to the “Results” section of the Abstract to include a summary of the outcomes for which we did not report a significant total effect estimate.

Purpose:

A 22-month household-based counseling intervention in Honduras was designed to improve practices, knowledge, and attitudes about maternal, neonatal, and child health.

Sample:

The eligibles were residents in 238 villages in one area, of which 176 were selected on the basis of “population size, geographic diversity, accessibility, and safety.”

The effect of this selection on generalisability was not assessed by analysing differences between the selected and not selected villages.

[Please address generalisability issues]

Response: As elaborated on in the study protocol, our preparation for this trial included geographic mapping of the 238 villages in the study region, during which we collected data on terrain, rainfall, distances to health facilities, accessibility, and safety. Of these villages, 176 were ultimately chosen to match into an 8x2 factorial randomization scheme. Excluded villages were either too small in size, not contributing to geographic diversity, poorly accessible for our Community Change Agents (CCAs), or unsafe. Because the reasons for exclusion inherently limited our ability to collect more extensive data about excluded villages (eg, villages that were not deemed safe for travel by our implementation team were not visited), formal analysis of the differences with intent to quantify generalizability was not possible. We have added text to emphasize this important implication for our study’s generalizability in the fourth paragraph of the “Discussion” section.

The final sample comprised 16,301 people in 5,633 households in 99 villages in the Copán region of Honduras.

In each village a proportion of households (0, or 0.05, 0.1, 0.2, 0.3, 0.5, 0.75, 1) were randomly assigned to the intervention. Then households were randomly assigned to the intervention according to the village’s dosage with covariate-constrained randomization to balance treatment proportions at the village and household levels. This strategy was effective as shown in Table 1 with baseline variables very similar between the Pure Control, the Random assignment within cluster control and the Targeted intervention groups.

Outcome measures:

The survey instrument included validated scales and internally developed items based on literature review, expert consult, Ministry of Health of Honduras advice and qualitative research with local

residents and village leaders.

A detailed analysis of the validity and reliability of the survey instruments was not provided.

Response: Our team worked extensively to develop a survey instrument to capture the wide range of outcomes of interest for the current study. This instrument was primarily composed of validated scales used widely to measure items related to reproductive, maternal, neonatal, and child health (RMNCH) outcomes. We conducted an extensive review of the RMNCH literature and consulted global RMNCH experts for their advice on the inclusion of appropriate items in the survey. We also did extensive formative research, including detailed qualitative individual interviews and focus groups, and cognitive interviewing to assess our survey's cultural relevance and consider regional language variations specific to the study area. This included a three-phase pilot study (see "Pilot work" section within supplementary appendix), in which survey questions were sequentially revised based on analyses showing lack of response variation or response patterns suggesting bias or misunderstanding. We believe the specific details of the analysis of the validity and reliability of each section of the instrument is beyond the scope of the manuscript, however, we have added text to our supplemental appendix within the "Pilot work" section detailing several of the validated scales that were included in our instrument.

Analysis:

The analysis was intention-to-treat. Individual-level data were analysed by multivariate logistic regression with binary outcomes. How much data was lost by making outcomes binary was not stated.

Response: Our statistical analysis plan at the outset included the use of binary outcomes to assess trial efficacy. The strengths of this approach included producing a more interpretable analysis (given our wide range and large number of outcomes) and also included generating results that could be compared against other, similar evidence-based interventions, the majority of which treat outcomes as binary (eg, Alive and Thrive trial cited in "Discussion" section which assessed exclusive breastfeeding). Many of our survey instruments elicited categorical responses which could easily be re-interpreted in a binary format. Most generally, only the minority of our survey questions elicited data that was continuous, so we do not believe that significant data was lost in this approach; however, future work which more closely investigates nuances in individual outcomes (especially those related to knowledges and attitudes) using more granular scales would contribute significant value to the current study. We hope this is acceptable.

One goal was to assess the average effect across all village-level treatment dosages and responses were pooled across all treatment arms and sampling weights were added to adjust for differential probabilities of treatment assignment. Clustered heteroskedasticity-robust standard errors that allowed for intragroup correlation were estimated at the village level.

Results:

Of the sample of 16,301, 13,881 completed baseline surveys and 10,263 completed the final survey 24 months later. Of the 6,038 individuals lost to the final survey, 50% migrated, 32% could not be reached, 12% refused to continue the study, and 4% died. There was no analysis of differences between completers and non-completers and effects on generalisability.
[please comment on generalisability]

Response: We agree that additional analysis of differences in attrition would be a helpful addition to the manuscript and have included an additional supplementary table, Table S10, in the appendix reporting these results. In summary, those who were lost to follow-up were more likely to be younger,

male, have a primary school or greater education, be single, not identify as Maya Chorti, and have greater self-rated physical and mental health. Importantly, there were no differences in attrition between targeted and untargeted households or between households of different wealth index quintiles. Text has been added to the first paragraph of the “Results” section and the fourth paragraph of the “Discussion” section.

Of the total 16,301 individuals across 5,633 households who were randomized, 77 villages with 4,410 households were randomized to the random targeting strategy and 22 villages with 1,223 households were randomized to the control arm with no treated households. Among the 77 villages assigned to the random targeting strategy, 11 villages were randomly assigned to each treatment dosage, or proportion of households targeted. Across all treatment dosages, a total of 1,815 households with 5,305 individuals were randomized to receive the intervention. Participant numbers varied widely from 48 to 571 between villages.

Three outcomes had statistically significant outcomes, with very wide 95% CIs. There was a 16.4% (95% CI 3.1%–29.8%, $p=0.016$) increase in the probability that parents had their newborn’s health checked by a professional in a health facility within three days of birth; a 19.6% (95% CI 4.2%–35.1%, $p=0.013$) increase in the probability that parents did not wrap a fajero around the umbilical cord in the first week after birth; and an 8.9% (95% CI 0.3%–17.5%, $p=0.043$) increase in the probability that the mother breastfed immediately after birth. [please comment on the reasons for the very wide 95% CIs and implications for the ability to generalise the results of the implementation of your study]

Response: The reasons for the wide confidence intervals reported in the current study are multifold. First, we used an intention-to-treat analysis design, which included all randomized respondents with eligible survey responses for each outcome, regardless of intervention receipt. As we report in the “Results - Quality assessment of intervention delivery” section, most (67%), but not all, households received modules for all 15 intervention topics. Additionally, although our study was sufficiently powered to detect significant effects, our sample size for practice outcomes was limited to parents who had a child after the delivery intervention period, which widened the confidence interval for primary outcomes relative to secondary outcomes. Finally, the contributions of both heterogeneity and ceiling effects as mentioned in the “Discussion” section may both also explain wide confidence intervals for our estimates. Overall, however, we believe that the confidence interval size alone does not have implications for the generalizability of our study due to the overall large power of the trial.

However, there were no significant effects of the intervention for seven other outcomes: knowledge or attitudes related to seeking first trimester prenatal care, facility-based birth; majority of the pregnancy danger signs; majority of the postnatal and newborn danger signs; immediate breastfeeding; father waiting at birth location or caring for children when infants were sick; prevention of diarrheal illness; or respiratory illness prevention and danger signs. No explanations for the non-significance of these outcomes were discussed.

[please discuss these hypotheses which were not supported and reasons why not]

Response: In response to several above comments related to limitations of the study, we have included edits to our discussion of limitations which also address potential reasons for the non-significance of the aforementioned outcomes (most notably, paragraphs 4, 7, and 8 of the “Discussion” section).

One potential reason might be that some knowledge and practices targeted by the intervention package were already present at high rates in our study area—we provide examples based on results from our baseline survey, including immediate breastfeeding (86%), avoiding harmful substances around umbilical cords (83%), and keeping the newborn warm and clothed after birth (99%).

Regardless of our intervention’s quality and potential impact, these ceiling effects may limit room for improvement. (These ceiling effects are encouraging and reflect progress previously made in Honduras towards goals for improving maternal and child health!).

Additionally, we discuss a theory on which our parent trial is partially based: that behavior change at the community level is preceded by the spread of changes in knowledge and attitudes. The results of the current study corroborate this theory, as we see a greater impact in secondary outcomes compared to primary outcomes. Finally, while our intervention package was aimed at impacting demand-side conditions, many practices also rely on supply-side conditions such as the accessibility of healthcare resources or quality of local health centers. Examples of such practices that were outcomes in the current study include facility-based births, preventative check-ups, care-seeking from medical professionals, and zinc treatment for diarrheal illness.

Reviewer: 1
Competing interests of Reviewer: I have no competing interests

Reviewer: 2
Competing interests of Reviewer: none

Reviewer: 3
Competing interests of Reviewer: None

Reviewer: 4
Competing interests of Reviewer: none

VERSION 2 – REVIEW

REVIEWER	Nash, Denis City University of New York System, Epidemiology and Biostatistics
REVIEW RETURNED	17-Oct-2023

GENERAL COMMENTS	The authors have done a good job addressing some of the issues raised by the peer review.
---

REVIEWER	Thomas, Roger University of Calgary, Family Medicine
REVIEW RETURNED	29-Sep-2023

GENERAL COMMENTS	This is a complex rural intervention for a very meritorious purpose. The four reviewers and the editor have identified all the methodological issues that arise in this complex situation and the authors have provided reasons for the intervention and analytic strategies they adopted. To provide maximum benefit for future researchers planning similar interventions in difficult to reach populations I advise that the editor ask the authors to provide in a box the key problems the reviewers and editor highlighted and the extent to which the authors were able to provide answers about intervention and analysis strategies. Their concise advice to future researchers would be valuable.
---